# Asserting a Functional Neurological Symptom Disorder with a Complementary Diagnostic Approach: A Brief Report

**DOI:** 10.3390/children10101601

**Published:** 2023-09-25

**Authors:** Anais Ogrizek, Thomas Ros, Maude Ludot, Marie-Rose Moro, Yves Hatchuel, Nicolas Garofalo Gomez, Rahmeth Radjack, Arthur Felix

**Affiliations:** 1Department of Adult and Child Psychiatry, Martinique University Hospital, F-97200 Fort-de-France, France; thomas.ros@chu-martinique.fr; 2Department of Child Psychiatry, University of Paris, Hôpital Cochin, Maison de Solenn, F-75014 Paris, France; maude.ludot@aphp.fr (M.L.); marie-rose.moro@aphp.fr (M.-R.M.); rahmeth.radjack@aphp.fr (R.R.); 3Department of Pediatrics, Martinique University Hospital, F-97200 Fort-de-France, France; yves.hatchuel@chu-martinique.fr (Y.H.); nicolas.garofalogomez@chu-martinique.fr (N.G.G.)

**Keywords:** functional neurological symptom disorder, Jonah complex, complementary approach

## Abstract

Introduction: Functional neurological symptom disorder (FNSD) is a common diagnosis among adolescents. However, we feel it is a difficult diagnosis to assess because of the diversity of its clinical manifestations, the rapid changes in its nosography over the years, and its common imbrication with established somatic diagnoses. We would like to illustrate this hypothesis through a case presentation and the original diagnostic process that emerged from it. Methods: We chose to present our diagnosis approach through the case of an 11-year-old boy who showed a progressive loss of motor and sensory function to the point of total dependency, and then suddenly switched between this state and a “normal” physical presentation, while deliriously claiming to be an angel. Results: All possible infectious, autoimmune, metabolic, and toxic disorders were ruled out. After the successive therapeutic failures of antidepressants and neuroleptics, FNSD was diagnosed. Conclusion: The DSM-5-TR classification was insufficient to explain the full clinical picture and a complementary approach (biblical, psychoanalytical, and historical) was used to analyze the cause of this atypical presentation.

## 1. Introduction

Conversion disorders, referred to as hysterical neurosis conversion type in earlier editions of the Diagnostic and Statistical Manual [1], were assigned the name of functional neurological symptom disorders in the latest (5th) edition, which classified them as part of the “Somatic symptom and related disorders.” These all share a common feature: “the prominence of somatic symptoms associated with significant distress and impairment” [2]. The somatic expression of psychological disorders are one of the leading motives of consultations in adolescent medicine [3]. FNSD is multifactorial and multidisciplinary but with a diagnosis most often at the interface between neurology and psychiatry [4].

Many labels and etiological theories, varying between cultures, have been attached to these disorders throughout the ages. First referred to as “hysteria” in Greek antiquity, this disorder was mainly characterized by the onset of nonepileptic seizures. It was believed that the god Asclepios could heal patients with hysteria by communicating with them during their dreams. Special temples were constructed across Greece, where priests would accompany patients into the land of dreams.

The Marquis de Puységur later theorized this phase of induced dreaming. Inspired by the practice of Mesmer, who used magnets to induce hysterical attacks artificially to heal patients [5], the Marquis was able to induce an altered state of consciousness that he called “magnetic somnambulism”, a state close to what is currently known as hypnosis during which the patient could communicate with the therapist. Based on his experience with his first patients, the Marquis de Puységur believed that during this state patients were very aware of their disease and could inform the doctor of how to treat them (Edelman, 2008). The famous French neurologist Charcot later identified this somnambulism state (or delirium) as the fourth stage of all hysterical seizures or attacks, which follows the first epileptoid stage (contortions and acrobatic poses), the second stage (clownism), and the third stage (emotional attitudes (“attitudes passionnelles”) and hallucinations). He considered the patient’s statements during this stage to be delirious and rejected the theory that suggestion or malingering could induce this state [6]. Starting with the DSM III, the symptoms related to this phase were excluded from the clinical description of hysteria and moved to a separate part called “dissociative syndrome”. Charcot also noted similarities in the symptoms, especially the recurrence of episodes of musculoskeletal contortions and modified states of consciousness, between patients with hysteria and those presenting demonic possession. Therefore, he classified “demonomania” as a form of hysteria. Many psychiatrists, including Hippolyte Bernheim and Janet, later argued that hysteria had a psychological etiology and that an attack of hysteria could therefore be induced—just as they could be cured—by suggestion [7].

In his book *Studies on Hysteria*, Freud assumed that this attack arose from repressed traumatic ideas or memories in the unconscious that were “converted” into physical symptoms. It would cease only if these ideas came freely to the patient’s conscious awareness, brought by the patient without any suggestion by the therapist [8]. This led Freud to invent the psychoanalytic couch as a device to prevent therapists from making any suggestions to their patients.

The history of mental suffering through bodily expression and somatic symptoms has focused on females, with hysteria derived from the Greek term “hystera” meaning “uterus”. Nonetheless, these symptoms currently appear to be prevalent among adolescents [3,9,10,11] as well as migrants [12,13,14]. It is difficult to quantify this illness, given the plethora of names for these disorders used by the medical community: somatoform disorders, conversion disorder, hysterical attack, psychogenic nonepileptic seizure, etc. This confusion results from the nosographic history over the years, as described above, the variety of these disorders’ clinical expressions, and sometimes their cultural interpretations, as well as the imbrication with verified somatic disorders that they share [11].

A recent literature review [11] identified potential risk factors for developing such a disorder. These include (but are not limited to) experience of trauma (mostly sexual or emotional) [15], adverse social or family environment (school bullying or family neglect) [16], or neurological predisposition [17,18,19]. Many articles also outline some specificities of functional neurological symptom disorders according to the patient’s cultural background with cultural etiologies and a potential “group effect” of the symptoms spreading within a defined group of adolescents [20]. Some research groups have proposed neurobiological models to explain FNSD: a dissociative brain process triggered by reflexive cortisol arousal or abnormal functional connectivity between the limbic structures and the supplementary and higher activity in the right amygdala, left anterior insula, and bilateral posterior cingulate [21].

The international community agrees on the beneficial effect of cognitive behavioral therapy and family therapy [22] for FNSD clinical presentations. Some therapists also argue that hypnosis therapy may be beneficial precisely because hypnosis may be an induced form of functional neurological disorder [23,24].

In view of the various clinical expressions of this diagnosis, its changing nosography over time, and its very common overlap with verified somatic diagnoses, as well as with cultural beliefs and interpretations, we think that the diagnostic approach to a patient presenting such symptoms cannot be restricted to the DSM-5-TR classification alone. We would like to illustrate this hypothesis through a case presentation and the original diagnostic process that emerged from it.

## 2. Materials and Methods

We describe the case of an 11-year-old patient admitted to the general pediatric ward and initially referred on suspicion of an acute/sub-acute encephalitis (infectious, autoimmune, or metabolic). The initial somatic work-up included the study of the cerebrospinal fluid (CSF) that was strictly normal for biochemistry, hematology, and viral/bacterial/fungal infection. There was also no sign of an active infectious event explaining the symptoms (the patient tested negative for Enterovirus, influenza, measles, mumps, Epstein–Barr virus, Cytomegalovirus, HHV6, and HIV). There was no overexpression of interferon genes in the CSF either. Metabolic investigations in blood, urine, and CSF (chromatography of urinary and blood organic amino acids, search for enzymatic deficiency, pyruvate/lactate cycle, glycosaminoglycan, and acyl carnitine) were normal. The autoimmune tests for blood autoantibodies or in the CSF (anti-nuclear antibody, autoimmune encephalitis antibody panel including MOG, Aquaporin 4, CASPR2, anti NMDAr, and GAD) were negative. Tests for a toxic cause (copper, lead, vitamin, or exogenous) were negative as well. The immune tests and study of the complement were normal. The ophthalmologic and Otorhinolaryngology work-up carried out showed no notable organic anomaly. The brain and spinal cord MRI were perfectly normal. Several electroencephalograms were performed including during sleep and over a long duration with video recording, but no seizures or abnormalities of the tracing were recorded. All the investigations carried out are summarized in the Appendix A. The family gave their consent to participate and for publication. The Institutional Review Board of the University Hospital of Martinique approved this study under number 2023/005.

## 3. Results

Patient information

Obadiah is an 11-year-old boy from Saint Lucia, a Caribbean island. His parents have been a couple for 18 years but have been always living apart. Although his mother has three other boys, all adopted, he is his parents’ only child. Obadiah is baptized. His name means “servant of God”. He is a social and cheerful boy and dreams of becoming a musician. He is in the sixth year of school and has good grades. No history of school bullying is reported. Obadiah’s mother was initially pregnant with twins: Obadiah and a little girl, who died in utero 3 days before delivery. The cause of death was unknown as it was detected in a follow-up ultrasonography without any symptoms. At the request of the family, no autopsy was carried out after delivery. Obadiah’s mother went through postpartum depression, for which she received medication and psychological counseling. At around 6 years old, shortly after being told about his twin sister’s death, Obadiah started developing phobias of water, cemeteries, funerals, and the dark. The case history up to Obadiah’s transfer to Martinique University Hospital can be found in Figure 1.

Clinical findings

At physical examination upon his transfer to the pediatric unit of the Martinique University Hospital, at the age of 11, Obadiah showed no reaction to sensorial stimuli: no speech but only grunts, no response to simple orders, and no active mobilization, with bilateral corticospinal tract dysfunction and horizontal nystagmus. He was fed via a gastric tube and urinated and defecated in a diaper. The fever was not confirmed during hospitalization.

Diagnostic assessment

Acute/subacute encephalitis was initially suspected. The explorations included a study of the cerebrospinal fluid, several blood and urine tests, brain and spinal MRIs, and video electroencephalography. This work-up found no infectious, toxic, metabolic, or autoimmune disorders.

On a somatic level: He did not show any complications from the long bed rest. As nurses followed the protocol for mobilizing patients, the child’s positions that were observed did not concur, which led us to believe that he might have changed position by himself when left alone. The partial regression of neurological symptoms was noted early during the stay. All medications were stopped and the neurological improvement continued throughout the stay.

On a psychiatric level: Fourteen days after the beginning of his hospitalization in Martinique, and shortly after the arrival of his father from Saint Lucia, Obadiah presented a brutal and total transformation. He presented himself as a walking and talking boy, explaining that his name was Xzavia and that he was an angel who had taken possession of Obadiah’s body to explain to us humans that he had been sent by God to save Obadiah from his illness. When asked what kind of illness Obadiah had, he would say that he suffered from autism and an abdominal illness. Xzavia would write lists of complaints and requests to “help Obadiah’s mother get her son well”. These lists included special foods Obadiah wanted to eat (mainly fast food) and things he wanted his mother to let him do (computer games, football, music, etc.). Xzavia appeared with a normal degree of consciousness and was well oriented in time and space. Although Xzavia showed appropriate behavior and no sign of neurological impairment, his thought content appeared delirious and around mystical themes. His report that he communicated with God led us to suspect intrapsychic hallucinations with mental automatism. He also said he could see God, which caused visual hallucinations. Xzavia could recall some events that had occurred while the patient presented as Obadiah and could report some of Obadiah’s thoughts. He was able to walk and eat normally. He had a normal appearance, with hospital clothing, good hygiene, and appropriate posture and gestures, although there was some doubt about possible facial mimicry. He showed appropriate behavior with good eye contact and appropriate body language and response to questions. Speech and language were normal in terms of volume, tone, rate, flow, and use of words. His attention was focused on the discussion, without interruption or any listening attitude. He showed good memory, easily recalling our last conversation and elements of Obadiah’s life story. His affect and mood were stable. His conviction that he was an angel sent by God to save Obadiah was unshakable. There were no elements of dissociation and the delirious process seemed quite logical and structured.

On the other hand, Obadiah had an encephalitis-like physical presentation. He had a constant upward gaze that made eye contact impossible. His head was hyper-extended. His left arm was hyper-flexed and he would grumble and show facial rictus whenever we tried to change his position. However, when he slept, his limbs were relaxed. He did not speak a word, could not walk, could not eat, was fed through a gastric tube, and urinated and defecated in a bucket. All communication with him was impossible.

The patient alternated, sometimes several times a day, between presenting as ‘Xzavia’, who could walk and communicate normally, and ‘Obadiah’, who had an encephalitis-like clinical picture and with whom any contact was impossible. Surprisingly, Xzavia could talk about some of the events that had occurred while Obadiah was in his encephalitis-like clinical presentation, and he would tell us some of Obadiah’s thoughts. Xzavia would only appear at the request of a few people, including Obadiah’s father, the hospital priest, and the hospital chaplain, but none of the doctors, nor Obadiah’s mother, whom Xzavia often rejected and insulted. The diagnosis of FNSD remained.

Therapeutic Intervention

None of the medication improved his clinical presentation (Table 1—Medication History). The family consulted the hospital chaplain almost every day. The father believed his son was possessed but the priest rejected that hypothesis due to Xzavia’s lack of biblical expertise, except for his precise knowledge of the story of Jonah, which he repeated frequently. After 48 days of hospitalization, Obadiah was discharged from Martinique University Hospital without any pharmaceutical treatment and transferred back to Saint Lucia.

Follow-Up and outcomes

He went back to St. Lucia where family therapy and individual hypnosis therapy were initiated, which resulted in a progressive improvement in his clinical presentation: Obadiah showed complete neurological recovery and the episodes of presenting as Xzavia ceased. He returned to school and rejoined his music band. He did not receive any pharmaceutical treatment during his follow-up. After a couple of months and after a follow-up 2 years later, recovery appeared complete.

## 4. Discussion

The diagnosis of FNSD was finally retained, as Obadiah’s clinical presentation met all the criteria in the DSM-5-TR description^2^. The symptoms appeared in a context of psychological stressors: the COVID-19 lockdown, family conflicts, the deterioration of school grades, the deprivation of extracurricular activities, admission to a highly desired new school, and the approach of the common entrance examination. Moreover, Obadiah obtained secondary gains from his medical condition: exclusive attention from his father and the reunification of his parents.

The DSM-5-TR FNSD diagnosis was nonetheless insufficient to explain the patient’s presentation as Xzavia. The initial hypothesis of a psychotic episode was rapidly abandoned after more acute knowledge of Obadiah’s clinical history and symptomatology, and the lack of improvement with neuroleptic treatment (Table 1). The rotation of the patient’s presentation between “Xzavia” and “Obadiah” as a dissociative symptom was discussed considering that dissociative amnesia can be comorbid to functional neurological symptom disorder. Nevertheless, no symptom of derealization nor amnesia were reported. Regarding the DSM-5-TR diagnostic criteria and the patient’s Afro-Caribbean cultural background, the diagnosis of “Other specified dissociative disorder’’ such as the “Chronic and recurrent syndromes of mixed dissociative symptoms” was also advanced, but the relationship that Xzavia and Obadiah seemed to maintain with one another led us to reject that hypothesis.

Moreover, the diagnosis of dissociative identity disorder was evoked but finally rejected considering the patient’s age and the lack of recurrent gaps in the recall of everyday events or important personal information.

Accordingly, we chose to explore old theories such as the Marquis de Puységur’s “magnetic somnambulism”, especially as neuroimaging has revealed similarities between the brain circuits involved in conversion disorders and the state of hypnosis [25]. As the induction of such a state is believed to be based on suggestibility, it was notable that Obadiah would switch to Xzavia in the presence of a person with great influence over him, such as his father, the hospital priest, or the hospital chaplain.

The Marquis de Puységur reported that “when placed in this state the patient can talk and present a different personality” and is “able to become his own doctor due to his exceptional lucidity regarding his disease” [26]. The patient could indicate to the doctor how to cure them and could reveal the real cause of their suffering. If this report is accurate, we could interpret Xzavia’s frequent mention of Jonah’s story as Obadiah’s way of expressing his deepest issues, the cause of the FNSD.

In the biblical story of the prophet Jonah [27], Jonah was sent by God to Nineveh to prophesize to its inhabitants that their town would be destroyed. Jonah tried to escape his holy mission by embarking on a boat in Jaffa in the direction of Tarshish and hiding in its cargo hold, deeply asleep. God’s rage on discovering Jonah’s disobedience provoked a storm that tossed Jonah out of the boat. He was then swallowed by a “great fish”, often depicted as a whale. He spent 3 days in the whale’s belly, begging for God’s forgiveness, which he obtained. He went back to Nineveh to accomplish his mission and announced the impending destruction of their city to its inhabitants. But, on discovering their town’s fate due to their own wickedness, Nineveh’s inhabitants became repentant. Seeing this, God finally decided to spare them. Jonah, feeling discredited, decided to retreat from the city to the nearby mountains and offered up his life.

Psychoanalysts have studied the story of Jonah since as early as 1912, when Carl Jung wrote his initial version of the analysis first published in English in 1955 (Jung, 1977). The story of Jonah’s engulfment by the whale calls attention to the great dangers of introspection and of searching for one’s self and one’s goal in life. Being vomited up from the whale’s belly might be considered a rebirth. Obadiah is a young adolescent, not yet even a teenager, facing difficult questions and trapped in his body. Showing physical regression might be a way to reflect on his future life goals. Erich Fromm, on the other hand, saw Jonah’s story as expressing this unwilling messenger’s determination to escape from all human life and regress to a fetal state in his mother’s belly, metaphorically presented first by the ship’s cargo hold and then by the whales belly [28]. Obadiah’s regressive clinical presentation could be assimilated to a fetal state and a desire to reconnect with the twin who died before they were born. Xzavia, an angel presented as his double, might be a reference to her. Joseph More hypothesized that Jonah might have perceived the inhabitants of Nineveh as his potential brothers and sisters. Terrified by fratricidal impulses potentially induced by the jealousy he felt seeing them receiving God’s love, Jonah might have decided to escape from Nineveh [29]. In Obadiah’s case, his mother recalled sibling rivalry between him and his brothers over their father’s love. From another perspective, Maslow (Maslow et al., 1993) refers to the Jonah complex and its syndrome as a fear of success that can lead individuals to self-sabotage. Prevalent in neurotic personalities who are also affected by conversion disorder [30], patients with this syndrome would tend to try to escape their destiny. Obadiah’s first symptoms occurred on the day he had finally entered the school of music he had dreamed of. In this context, the symptoms might be related to his fear of finally reaching his highest goal in life and failing.

## 5. Conclusions

The DSM-5-TR FNSD diagnosis was insufficient to explain the full clinical picture in such cases, and at times a more comprehensive/complementary approach may be more beneficial. Applying the principle of complementarity—in this case, history, religion (biblical studies), and psychoanalysis—recommended by Devereux [31] has led us to consider the story of Jonah repeatedly mentioned by Obadiah through his twin angel Xzavia as a potential key to explain the real issues that might have induced FNSD in this adolescent. Future research would be needed to study the efficacy of a complementary approach for this specific diagnosis, as it has already proven its efficacy in many other diagnoses [32,33,34].

## Figures and Tables

**Figure 1 children-10-01601-f001:**
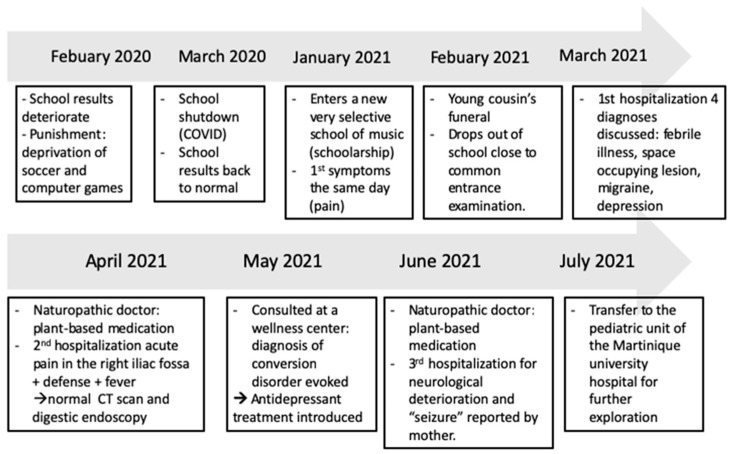
Case history up to transfer to Martinique University Hospital.

**Table 1 children-10-01601-t001:** Medication history.

Date	Time	Drug	Dose	Indication
15 January	1 week	Paracetamol	500 mg/6 h	Pain
6 April	1 week	Moringa (among others)		
19 May	2 months	Fluoxetine	20 mg/d	Depression
1 June	1 week	GABA, Melatonin, digestive enzymes, acidophilus, and mentat		
21 June + 9 August	1 day	Clonazepam	1 mg IR	“Seizure”
26 July	3 weeks	Clonazepam	9 drops/6 h	Post lumbar puncture syndrome
26 July	3 weeks	Hydroxyzine	75 mg/d	Post lumbar puncture syndrome
26 July	3 weeks	Amitriptyline	30 mg/day	Post lumbar puncture syndrome
9 August	3 weeks	Levetiracetam	750 mg/day	“Seizure”
16 August	3 weeks	Risperidone	1 mg/day	Delusions
25 August	3 weeks	Diazepam	15 mg/day	“Seizure”

## Data Availability

The data presented in this study are available in the article and Appendix A. Other data are available on request from the corresponding authors.

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
