# Peer review of "Asserting a Functional Neurological Symptom Disorder with a Complementary Diagnostic Approach: A Brief Report"

_children, 2023, doi:10.3390/children10101601_

Round 1
Reviewer 1 Report
Authors presented an interesting case report. Overall well written, however the case report may include few additional details as detailed below.
1) In the abstract, the introduction can be a bit more clear about what is going to be presented in this case, namely present the aim of the case report early on.
2) I would detail the specifics of biochemistry, hematology, viral/infectious, immune and endocrinological workup. May be presented as supplementary.
3) I would detail the nature of work up instead of just saying ophthalmologic and otorhinolaryngology work-up…
4) What was the death for little sisters’ in utero death?
5) How does the author explain fever with a somatic disorder?
6) Argument about belief that he may have changed position when left alone is odd. Does that mean in this hospital, there is no protocol to change patient’s position by nurse?
7) Line 158, no need to with SSRI in upper cases.
8) Line 162, “as himself” is a repetition.
9) Would recommend author to include “the fact that DSM-V classification was insufficient to explain the full clinical picture in such cases and at times a more comprehensive/complementary approach may be more beneficial.
10) Author can provide some future directions to study efficacy of complementary approach in such cases.
Author Response
Authors presented an interesting case report. Overall well written, however the case report may include few additional details as detailed below.
We would first like to thank you for your interest in our article and for your relevant comments on our work. We have made changes as recommended and would like to resubmit this article for your assessment.
- In the abstract, the introduction can be a bit more clear about what is going to be presented in this case, namely present the aim of the case report early on.
Thank you very much for your comment. We have added relevant information in the introduction on our clinical approach and the aim of the case report presentation.
Modification in the text:
L 29-32: “However, we feel it is a difficult diagnosis to assess because of the diversity of its clinical manifestations, the rapid changes in its nosography over the years, and its common imbrication with established somatic diagnoses. We would like to illustrate this hypothesis through a case presentation and the original diagnostic process that emerged from it. »
2) I would detail the specifics of biochemistry, hematology, viral/infectious, immune and endocrinological workup. May be presented as supplementary.
Thanks for your relevant comment, we have added a supplementary table (supp table 1) detailing the carried-out explorations.
3) I would detail the nature of work up instead of just saying ophthalmologic and otorhinolaryngology work-up…
Thanks for your comment, we have detailed the work-up performed in supplementary table.
4) What was the death for little sisters’ in utero death?
The cause of the death is unknown. It was detected in a follow-up ultrasonography and no autopsy was carried out.
Modification in the text:
L 147 -149 « The cause of the death was unknown as it was detected in a follow-up ultrasonography without any symptoms. At the request of the family, no autopsy was carried out after delivery.
5) How does the author explain fever with a somatic disorder?
The fever was only reported by the parents but never confirmed in the hospital. They reported they had not measured the temperature, but felt that Obadiah was warm. A sentence was added in the text to clarify this point.
Modification in the text:
L 162-163: “The fever was not confirmed during hospitalization.”
6) Argument about belief that he may have changed position when left alone is odd. Does that mean in this hospital, there is no protocol to change patient’s position by nurse?
Thank you for your very relevant comment. There is indeed a unit protocol for mobilizing patients, but almost each and every time the nurse came to change the child's position, she reported his position was different from the previous indicated by the protocol. This led us to think that he had been moving by himself. A sentence was added to clarify this point
Modification in the text:
L 169-172: “As nurses followed the protocol for mobilizing patients, the child's positions that were observed did not concur, which led us to believe that he might have changed position by himself when left alone.”
7) Line 158, no need to with SSRI in upper cases.
The sentenced was removed as asked.
8) Line 162, “as himself” is a repetition.
The repetition was removed as asked.
9) Would recommend author to include “the fact that DSM-V classification was insufficient to explain the full clinical picture in such cases and at times a more comprehensive/complementary approach may be more beneficial.
Thanks for your comment, the sentence was added to the conclusion as asked.
Modification in the text:
L 302_304: “The DSM-5-TR FNSD diagnosis was insufficient to explain the full clinical picture in such cases and at times a more comprehensive/complementary approach may be more beneficial”.
10) Author can provide some future directions to study efficacy of complementary approach in such cases.
Related to your comments we have added a sentence regarding the need for future research on the efficacy of complementary approach regarding this specific diagnosis.
Modification in the text:
L307_309: Future research would be needed to study the efficacy of complementary approach for this specific diagnosis like it has already proven efficacy in many other diagnosis
Reviewer 2 Report
This brief report presents an atypical case of FNSD and questions about a more integrated approach to its diagnosis. It leaves the clinical reader with questions about the more correct diagnostic framework. Overall, the manuscript is well written and quite interesting. I really liked the historical insights within the introduction. However, I'd like to read more about articles and reviews from scientific literature in the introduction, in order to make the paper sound more scientific. Maybe something more about differential diagnoses with FNSD could also be added, especially in the discussion section, where much space is left to the historical-psychoanalytic component, however very interesting.
I also suggest to modify "DSM-V" with the correct name "DSM-5". Anyhow, you could definitely mention DSM-5-TR instead of DSM-5, since it is the latest version of the manual.
The present manuscript is written in an understandable and sufficiently clear way. Please double check your English for minor typos (for example page 1, line 29; page 4, line 162).
Author Response
This brief report presents an atypical case of FNSD and questions about a more integrated approach to its diagnosis. It leaves the clinical reader with questions about the more correct diagnostic framework. Overall, the manuscript is well written and quite interesting. I really liked the historical insights within the introduction.
We would first like to thank you for your interest in our article and for your relevant comments on our work. We have made changes as recommended and would like to resubmit this article for your assessment.
However, I'd like to read more about articles and reviews from scientific literature in the introduction, in order to make the paper sound more scientific.
As asked, we have added text and scientific literature references in the introduction, especially regarding research results on possible entanglements between neurological origin and psychiatric expression.
Modification in the text:
- L 50_52: “Somatic expression of psychological disorders are one of the leading motives of consultations in adolescent medicine; FNSD is multifactorial and multidisciplinary but with a diagnosis most often at the interface between neurology and psychiatry
- L 101_104 : “Some research groups have proposed neurobiological models to explain FNSD: dissociative brain process triggered by reflexive cortisol arousal or abdnormal functional connectivity between the limbic structures and the supplementary and higher activity in the right amygdala, left anterior insula and bilateral posterior cingulate
Maybe something more about differential diagnoses with FNSD could also be added, especially in the discussion section, where much space is left to the historical-psychoanalytic component, however very interesting.
Thank you for your comment. In response we have added a whole paragraph to summarize our clinical discussions and process regarding this patient’s diagnosis. We hope this will meet your expectations.
Modification in the text:
L 240-253 : “The initial hypothesis of a psychotic episode was rapidly abandoned after more acute knowledge of Obadiah’s clinical history and symptomatology, and the lack of improvement with neuroleptic treatment (Table 1).
The rotation of patient’s presentation between “Xzavia” and “Obadiah” as a dissociative symptom was discussed considering that dissociative amnesia can be comorbid to functional neurological symptom disorder. Nevertheless, no symptom of derealization nor amnesia were reported. As regard to the DSM-5-TR diagnostic criteria and the patient’s afro-Caribbean cultural background, the diagnosis of “Other specified dissociative disorder’’ such as the “Chronic and recurrent syndromes of mixed dissociative symptoms” was also advanced. But the relationship that Xzavia and Obadiah seemed to maintain with one another led us to reject that hypothesis.
Moreover, the diagnosis of dissociative identity disorder was evoked but finally rejected considering the patient’s age and the lack of recurrent gaps in the recall of everyday events or important personal information.
I also suggest to modify "DSM-V" with the correct name "DSM-5". Anyhow, you could definitely mention DSM-5-TR instead of DSM-5, since it is the latest version of the manual.
Thanks for your relevant comment. DSM-V was changed to DSM-5-TR as recommended.
Comments on the Quality of English Language
The present manuscript is written in an understandable and sufficiently clear way. Please double check your English for minor typos (for example page 1, line 29; page 4, line 162).
Thank you for your comment. The new manuscript has been critically and entirely reviewed by a native English speaker and the indicated errors were corrected in the text.
